# Sharing a River with Downstream Externalities

**Sarina Steinmann and Ralph Winkler \***

Department of Economics and Oeschger Centre for Climate Change Research, University of Bern, Schanzeneckstrasse 1, CH-3012 Bern, Switzerland; sarinasteinmann@gmail.com

**\*** Correspondence: mail@ralph-winkler.de

**Abstract:** We consider the problem of efficient emission abatement in a multi polluter setting, where agents are located along a river in which net emissions accumulate and induce negative externalities to downstream riparians. Assuming a cooperative transferable utility game, we seek welfare distributions that satisfy all agents' participation constraints and, in addition, a fairness constraint implying that no coalition of agents should be better off than it were if all non-members of the coalition would not pollute the river at all. We show that the downstream incremental distribution, as introduced by Ambec and Sprumont (2002), is the only welfare distribution satisfying both constraints. In addition, we show that this result holds true for numerous extensions of our model.

**Keywords:** downstream externalities; downstream incremental distribution; optimal emission abatement; river pollution

---

## 1. Introduction

Industries and cities all around the world have historically been concentrated along rivers, since rivers provide means of transportation, food production, energy generation and drinking water. Because of this intensive utilization, many rivers and streams have been and still are being heavily polluted. Excessive pollution worsens water quality, which reduces economic profits and negatively impacts wildlife and human health. One specific characteristic of rivers is that pollutants discharged into the river are carried downriver. As a consequence, it is the downstream riparians rather than the polluters themselves who bear the negative consequences of the emissions discharged into the river. Moreover, if upstream polluters and downstream riparians belong to different jurisdictions, polluters may have little incentive to abate their emissions, because they cannot be held liable for the pollution damage caused in other jurisdictions.

In this paper, we consider the problem of efficient emission abatement among agents located along a river, where upstream emissions cause negative externalities to all downstream agents. This setting can be characterized as a cooperative transferable utility game with two sources of externalities.[1] First, upstream emissions impose negative externalities on downstream agents. Second, cooperative behavior among a subset of agents (a so-called coalition) imposes positive externalities upon agents located in between different connected subsets of this coalition. Due to this second kind of externalities, the core is, in general, empty. As a consequence, we restrict our attention to the non-cooperative core ($\gamma$-core), i.e., the set of partitions which consists of one coalition and only singletons otherwise.

---

[1] As we characterize the game in cost space instead of utility space, one might rather speak of a "transferable cost" game.

The non-cooperative core imposes cost upper bounds for any coalition, which can be interpreted as a participation constraint that has to be satisfied by any cost distribution to be acceptable to all agents. In addition, we impose cost lower bounds, which are inspired by the aspiration welfare principle, i.e., no coalition of agents should have lower costs than it can secure for itself if all non-members of the coalition did not pollute the river at all. We show that the downstream incremental distribution, as introduced by Ambec and Sprumont [1], is the only distribution simultaneously satisfying the non-cooperative core upper cost bounds and the aspiration lower cost bounds.

Several other papers propose alternative sharing rules to the downstream incremental distribution in settings similar to the one proposed by [1]. Interpreting the river sharing problem as a line-graph game, Van den Brink et al. [2] derive four different efficient solutions including the downstream incremental distribution by imposing various properties with respect to deleting edges of the line-graph. However, they do not address fairness issues and consider non-satiable agents. Allowing for multiple springs and satiable agents with respect to water consumption, Van den Brink et al. [3] propose a class of weighted hierarchical welfare distributions based on the *Territorial Integration of all Basin States* (TIBS) principle from international water law, which includes the downstream incremental distribution as a special case. Ansink and Weikard [4] concentrate on reallocations of the resource itself instead of the reallocation of welfare by an appropriate transfer scheme. In case of water scarcity, the agents' overlapping claims to river water render it a contested resource similar to a bankruptcy problem. They propose a class of sequential sharing rules based on bankruptcy theory and compare them to other sharing rules, including the downstream incremental distribution. Demange [5] considers hierarchies without externalities and shows that the *hierarchical outcome* satisfies the core bounds for all connected coalitions for all super-additive cooperative games. However, the hierarchical outcome may violate core bounds for non-connected coalitions. If the hierarchy is a river, then the hierarchical outcome corresponds to the counterpart of the downstream incremental distribution.

The existing literature on transboundary pollution in river basins mainly focusses on the case of two jurisdictions. Notable exceptions include [6–8]. Ni and Wang [6] derive cooperative sharing rules for the costs of cleaning a river from two other principles of international water law: *Absolute Territorial Sovereignty* (ATS) claims that every jurisdiction has exclusive rights to use the water on its territory, while *Unlimited Territorial Integrity* (UTI) expands these exclusive use rights to all water originating within and upstream of a respective jurisdiction. They adapt these principles to the case of pollution responsibility and derive axioms characterizing the two resulting cost sharing principles. They also show that these cost-sharing principles correspond to the Shapley value solutions of the corresponding cost-sharing games. However, assuming exogenously given costs for cleaning the river, they are only concerned with the distribution of these costs. In contrast, pollution levels in our model are endogenously determined by the actions of the agents. Thus, we are concerned about finding cost sharing distributions that are acceptable to all agents and, at the same time, give incentives to choose efficient emission abatement levels in the first place.

In line with the literature on international environmental agreements, Gengenbach et al. [7] model river pollution as a two-stage-cartel-formation game. In the first stage, agents decide whether to join a coalition, while pollution abatement levels are chosen in the second stage. In the absence of a supranational authority, abatement levels are in general inefficiently low, as all agents have an incentive to free ride on the abatement efforts of their upstream neighbors. Analyzing the formation of stable coalitions they find that the location of agents has no impact on coalition stability but rather impacts on environmental outcomes. In contrast, we employ a cooperative game setting.

Our paper is most closely related to [1,8,9]. On the one hand, Ambec and Sprumont [1] and Ambec and Ehlers [9] apply an axiomatic cooperative game theoretic approach to the efficient sharing of water along a river basin. In ref. [1], agents derive strictly increasing benefits from water consumption, while in ref. [9] the results are generalized to agents exhibiting satiation in water consumption. They show that the downstream incremental distribution is the only welfare distribution satisfying the non-cooperative core lower bounds and the aspiration welfare upper bounds. Our paper shares

the axiomatic game theoretic approach and can be interpreted as a generalization of the results of [9] to commodities with public good properties. While water consumption is a purely private good, emission abatement exhibits public good characteristics, as it imposes positive externalities on all downstream agents. These additional externalities impose non-trivial complications for proving that the downstream incremental distribution satisfies the non-cooperative cost upper bounds and the aspiration cost lower bounds in the formulation of our river pollution model. Van der Laan and Moes [8], on the other hand, analyze a river pollution model similar to ours.[2] As described in detail in Section 3, it is not trivial to assign cost upper bounds to coalitions in the river pollution model, as these depend on the cooperation structure of all riparians outside the coalition. We follow [9] in restricting attention to the non-cooperative core in defining unique cost upper bounds for all possible coalitions. Together with cost lower bounds, which are inspired by the UTI doctrine, we seek cost distributions that satisfy both cost lower bounds and cost upper bounds for all feasible coalitions. Ref. [8] follow a different route. To each upstream coalition they assign a value equal to our cost upper bound, while each downstream coalition gets assigned a value equal to our cost lower bounds. To reconcile the problem that the sum of values of each upstream coalition and its counterpart downstream coalition exceed the total cooperation gain, they distribute the deficit among coalitions according to the $\alpha$-TIBS fairness principle, a generalization of component fairness (see [10]), which includes the downstream incremental distribution as a special case. Thus, ref. [8] characterize a class of welfare distributions derived by distributing the welfare deficit, which is implied by their assignment of values to coalitions, according to $\alpha$-TIBS fairness. We, in contrast, focus on the derivation of the unique sharing rule that satisfies the two axioms of lower and upper cost bounds.

## 2. A River Sharing Model with Downstream Pollution Externalities

Consider a set of agents $N = \{1, ..., n\}$, which are located along a river. Without loss of generality, agents are numbered from upstream to downstream, i.e., $i < j$ indicates that agent $j$ is located downriver of agent $i$. We follow [1,9] in defining the set of agents preceding agent $i$ by $P_i = \{1, ..., i\}$, with the strict predecessors of agent $i$ indicated as $P_i \backslash i = \{1, ..., i-1\}$. Analogously, the set of agents following agent $i$ is defined by $F_i = \{i, ..., n\}$, where $F_i \backslash i = \{i+1, ..., n\}$ denotes the set of agents strictly located downriver of agent $i$.

Each agent $i$ along the river produces gross emissions in exogenously given amount $e_i$. An agent $i$ may choose to abate the amount $x_i$ with $0 \leq x_i \leq e_i$, the costs of which are given by the strictly increasing, twice differentiable and strictly convex abatement cost function $c_i(x_i)$. Without loss of generality, we assume that not abating at all induces no abatement costs, i.e., $c_i(0) = 0$. Net emissions $e_i - x_i$ are passed into the river where they accumulate and are carried along its course. Assuming that net emissions of agent $i$ are discharged into the river after agent $i$'s but before agent $i+1$'s location, and that there is no pollution at the rivers' source, the ambient pollution level $q_i$ at the location of agent $i$ is given by the sum of net emissions of all strict predecessors of agent $i$:

$$q_i = \sum_{j \in P_i \backslash i} \gamma_{ji}(e_j - x_j), \quad \forall\, i \in N, \tag{1}$$

with $0 < \gamma_{ji} \leq 1$. $\gamma_{ji}$ represents the assimilative capacity of the river, i.e., what fraction of the net emissions released by agent $j$ actually reach agent $i$. As the vector of abatement efforts $x = (x_1, \ldots, x_n)$,

---

[2]    In ref. [8] agents choose *pollution levels* to maximize the difference between benefits from individual pollution levels and damages depending on the vector of pollution levels over all agents, while in our model framework agents choose *abatement levels* to minimize the sum of the costs of individual abatement and the damage from aggregated pollution (see Section 2). Our model set-up could also be expressed in terms of pollution levels, yielding a model structure formally identical to [8].

together with the vector of exogenously given emissions $e = (e_1, \ldots, e_n)$, fully determine the vector of ambient pollution levels, we shall often write the ambient pollution levels as a function of the vector $x$:

$$q(x) = \big(q_1(x), \ldots, q_n(x)\big). \tag{2}$$

The ambient pollution level $q_i$ causes damage costs to agent $i$, the amount of which is given by the increasing, twice differentiable and convex damage cost function $d_i(q_i)$. Thus, the net emissions $e_i - x_i$ released by agent $i$ induce negative externalities for all downriver agents $j > i$, but not for agent $i$ himself or all upstream agents $j < i$.

The total costs $k_i$ agent $i$ faces are the sum of abatement and damage costs:

$$k_i(x_i, q_i) = d_i(q_i) + c_i(x_i) . \tag{3}$$

A river sharing problem is characterized by $(N, e, c, d)$, where $c = (c_1, \ldots, c_n)$ and $d = (d_1, \ldots, d_n)$ denote the vectors of abatement and damage cost functions. Given a river sharing problem, the distribution of total costs $k_i$ among all agents $i$ is determined by the emission abatement allocation $x$. Our assumptions about the accumulation of emissions along the river, as described in the previous paragraph, imply the following proposition.

**Proposition 1** (No abatement is dominant strategy). *Given a river sharing problem $(N, e, c, d)$ it is a dominant strategy for all agents $i \in N$ not to abate at all, i.e., $x_i = 0$.*

**Proof.** The damage costs of agent $i$ only depend on $q_i$, which are not influenced by $x_i$. As costs $c_i$ are strictly increasing in the amount of emission abatement $x_i$, given $q_i$, total costs are minimized by setting $x_i = 0$. $\square$

Proposition 1 states that agents who only consider their own total costs will never abate. In particular, this implies that if the river sharing problem $(N, e, c, d)$ is considered to be a non-cooperative game among the agents $i \in N$, the unique Nash equilibrium is given by $\hat{x}_i = 0$ for all $i \in N$ (no matter whether agents are considered to decide sequentially or simultaneously). However, this outcome is, in general, inefficient. In particular, if we assume that money transfers between agents are possible and agents have unbounded resources for such transfers, the efficient emission abatement allocation $x^\star$ minimizes the sum of total costs $k_i$ among all agents. The following proposition establishes that such an allocation exists and is also unique.

**Proposition 2** (Existence and uniqueness of efficient allocation). *Given a river sharing problem $(N, e, c, d)$ there exists a unique vector $x^\star$, which is the solution to the following constrained minimization problem:*

$$\min_{\{x_i\}_{i=1}^n} \sum_{i=1}^n k_i\big(x_i, q_i(x)\big) \quad \text{subject to} \tag{4a}$$

$$q_i(x) = \sum_{j \in P_i \setminus i} \gamma_{ji}(e_j - x_j) , \quad \forall\, i \in N , \tag{4b}$$

$$0 \leq x_i \leq e_i , \quad \forall\, i \in N . \tag{4c}$$

**Proof.** Existence and uniqueness follow directly from the strict convexity of the total costs functions $k_i(x_i, q_i)$. $\square$

Let $t_i$ denote the money payments. We impose $\sum_{i=1}^n t_i = 0$ and define agent $i$'s after transfer costs $z_i$ as:

$$z_i = k_i(x_i, q_i) + t_i . \tag{5}$$

Obviously, any vector $z = (z_1, \ldots, z_n)$ with $\sum_{i=1}^n z_i = \sum_{i=1}^n k_i\big(x_i^\star, q_i(x^\star)\big)$ is an efficient cost distribution, as it implies a unique vector of transfer payments $t_i = z_i - k_i\big(x_i^\star, q_i(x^\star)\big)$ with $\sum_{i=1}^N t_i = 0$

(no waste of money) and achieves the cost minimum $\sum_{i=1}^{N} k_i(x_i^\star, q_i(x^\star))$. In the following, we call any efficient cost distribution a *river sharing agreement*. The main problem will be which one to choose among this infinite set.

## 3. Coalitions and Cost Upper Bounds

A non-empty subset of agents $S \subseteq N$ is called a coalition if the agents of $S$ choose their emission abatements such as to minimize the sum of total costs among all coalition members. Denoting by $minS$ and $maxS$ the most upstream, respectively the most downstream member of coalition $S$, the coalition $S$ is *connected* or *consecutive* if all agents $j$ with $minS < j < maxS$ are also members of the coalition $S$.

Given a coalition $S$ and a vector of abatement levels $x^{N \setminus S} = \{x_i\}_{i \notin S}$ of all agents $i \notin S$, we define the secure costs $v(S, x^{N \setminus S})$ of a coalition $S$ as the minimum value of the sum of the total costs $k_i$ over all members of the coalition:

$$v(S, x^{N \setminus S}) = \sum_{i \in S} k_i\left(x_i^v(S, x^{N \setminus S}), q_i(x^v(S, x^{N \setminus S}))\right), \tag{6}$$

where $x^v(S, x^{N \setminus S}) = \left(x_1^v(S, x^{N \setminus S}), \ldots, x_n^v(S, x^{N \setminus S})\right)$ denotes the solution to

$$\min_{\{x_i\}_{i \in S}} \sum_{i \in S} k_i\left(x_i, q_i(x)\right) \quad \text{subject to} \tag{7a}$$

$$q_i(x) = \sum_{j \in P_i \setminus i} \gamma_{ji}(e_j - x_j), \quad \forall i \in N, \tag{7b}$$

$$0 \leq x_i \leq e_i, \quad \forall i \in S, \tag{7c}$$

$$x^{N \setminus S} \text{ given}. \tag{7d}$$

It is obvious from the above definition that both the allocation of abatement efforts $x^v(S, x^{N \setminus S})$ and the secure costs $v(S, x^{N \setminus S})$ of the coalition $S$ depend, in general, on the behavior of the agents not belonging to the coalition $S$. As an example, consider the coalition $S = \{k, \ldots, n\}$. In particular the pollution level $q_k$ (but also the pollution levels $q_i$ with $i > k$) depends on the amount of emission abatement undertaken by the agents $i$ with $i < k$. According to Proposition 1, if these agents $i < k$ only minimize their own sum of abatement and damage costs, they would not abate at all, implying a pollution level of $q_k = \sum_{j \in P_k \setminus k} \gamma_{jk} e_j$. If however, the agents 1 to $k-1$ form a coalition $T$ and minimize their joint total costs, they will, in general, choose $x_j > 0$ for at least some $j \in 1, \ldots, k-1$. This implies a pollution level of $q_k < \sum_{j \in P_k \setminus k} \gamma_{jk} e_j$ which reduces the minimal costs $v(S, x^{N \setminus S})$ coalition $S$ can secure for itself. Thus, analogously to [9], cooperation exerts a positive externality on the coalition $S$.

In the following, we restrict our attention to the *non-cooperative core*, i.e., we assume that all non-members of a coalition $S$ behave non-cooperatively, which according to Proposition 1 implies that they do not abate at all.[3] Then, condition (7d) is replaced by $x_j = 0$ for all $j \notin S$, and the secure costs $v(S, x^{N \setminus S})$ of a coalition $S$ are well defined and unique (as the resulting optimization problem is a subproblem of the one analyzed in Proposition 2). The reason is like in [9]: the structure of the river sharing problem $(N, e, c, d)$, as described in detail in Section 2, is such that only the non-cooperative core is guaranteed to be non-empty. In the following, we denote the secure costs of the non-cooperative core by $v(S)$, which satisfy the following condition:

---

[3]　This concept of $\gamma$-*core* was first derived by [11]. Note that in our particular model set-up, in which emission abatement choices of $x_i = 0$ are dominant strategies for all agents $i \notin S$, the $\gamma$-core is also identical to the $\alpha$-*core* (where agents outside the coalition are assumed to react to the strategy of the coalition such as to minimize the pay-off of the coalition $S$) and the $\beta$-*core* (where agents outside the coalition $S$ are assumed to move first and choose emissions such as to minimize the maximum pay-off of coalition $S$), see [12,13].

**Proposition 3.** *For any $T \subset N$ with $\min T > m_j$ and any $m_j \in N$ the following inequality holds:*

$$v(P_{m_j} \cup T) - v(P_{m_j} \setminus m_j \cup T) \leq v(P_{m_j}) - v(P_{m_j} \setminus m_j) \, . \tag{8}$$

The proof of Proposition 3 is given in the Appendix A. Proposition 3 says that the increase of the cost upper bound is weakly higher if agent $m_j$ joins a coalition of all his predecessors compared to a coalition of all his predecessors and some other agents located more downstream to agent $m_j$. Thus, a member $m_j$ is weakly more valuable to a coalition that has members downstream of agent $m_j$. The reason is straight forward: Abatement at the location of member $m_j$ can only benefit the coalition if it exhibits members that are located downstream of agent $m_j$.

Like [9], we impose the secure costs as the participation constraint of any coalition $S$. A coalition $S$ will only agree to a river sharing agreement if it is not worse off than without the agreement. Thus, a river sharing agreement should at most assign the secure costs $v(S)$ to any coalition $S$, as otherwise the coalition would block the agreement knowing that it can achieve at least $v(S)$ on its own. Hence, $v(S)$ defines cost upper bounds for any coalition $S$ a river sharing agreement must satisfy in order not to be blocked.

## 4. Cost Lower Bounds

Ref. [9] also impose welfare upper bounds that are inspired by the *unlimited territorial integrity* (UTI) doctrine. In case of water consumption, UTI claims that all agents are entitled to consume the full stream of water originating upstream from their location and, thus, have a legitimate claim to the corresponding welfare level such a consumption generates. As such claims are, in general, incompatible if water is scarce, refs. [1,9] interpret them as welfare upper bounds agents may legitimately aspire to.

The straightforward translation of these aspiration welfare upper bounds to the case of our river pollution model is to define the minimal costs a coalition $S$ can ensure if all non-members of the coalition would abate all their emissions, and thus, not pollute the river at all. Formally, these cost lower bounds $a(S)$ are given by:

$$a(S) = \sum_{i \in S} k_i \left( x_i^a(S), q_i(x^a(S)) \right) , \tag{9}$$

where $x^a(S) = \left( x_1^a(S), \ldots, x_n^a(S) \right)$ denotes the solution to

$$\min_{\{x_i\}_{i \in S}} \sum_{i \in S} k_i(x_i, q_i(x)) \quad \text{subject to} \tag{10a}$$

$$q_i(x) = \sum_{j \in P_i \setminus i} \gamma_{ji}(e_j - x_j) , \quad \forall i \in N , \tag{10b}$$

$$0 \leq x_i \leq e_i , \quad \forall i \in S , \tag{10c}$$

$$x_j = e_j , \quad \forall j \notin S . \tag{10d}$$

The cost lower bounds $a(S)$ can be interpreted as a fairness condition: no coalition $S$ should enjoy lower costs than the costs it could secure itself if all non-members of the coalition would not pollute the river at all. In addition, the cost upper bounds $a(S)$ satisfy the following condition:

**Proposition 4.** *For any $S \subset T \subseteq N \setminus i$, the following inequality holds:*

$$a(S \cup i) - a(S) \leq a(T \cup i) - a(T). \tag{11}$$

We prove Proposition 4 in Appendix B. Proposition 4 establishes that $a(s)$ is convex. That is, the incremental value of the membership of agent $i$ to a coalition is weakly increasing in the coalition size.

## 5. The Downstream Incremental Distribution

Like in refs. [1,9], there is a connection between the non-cooperative core upper bounds $v(S)$ and the cost lower bounds $a(S)$: For the coalition of all predecessors of agent $i$ they coincide, i.e., $v(P_i) = a(P_i)$. Thus, for any coalition of predecessors $P_i$ it is clear that the only river sharing agreement satisfying both the cost upper and cost lower bounds is the so called *downstream incremental distribution* (*DID*) defined by

$$z_i^\star = v(P_i) - v(P_i \backslash i) = a(P_i) - a(P_i \backslash i) , \quad \forall\, i \in N . \tag{12}$$

The *DID* assigns every agent his marginal contribution to the coalition composed of his predecessors along the river. As a consequence, the *DID* is the only candidate for a river sharing agreement that at the same time satisfies the non-cooperative core upper bounds $v(S)$ and the cost lower bounds $a(S)$ for any coalition $S$. The following theorem establishes that the *DID*, in fact, satisfies the non-cooperative core upper bounds $v(S)$ and the cost lower bounds $a(S)$ for any coalition $S$.

**Theorem 1** (Only *DID* satisfies cost upper and lower bounds)**.** *The downstream incremental distribution (DID) $z^\star$ is the only river sharing agreement satisfying the non-cooperative core upper bounds $v(S)$ and the cost lower bounds $a(S)$ for any coalition S.*

**Proof.** The proof is split into three parts. In the first part, we show that the *DID* satisfies the non-cooperative core upper bounds for any coalition *S*. In part two, we prove that the *DID* also satisfies the cost lower bounds for any coalition *S* and, finally, in the third part, we show that any river sharing agreement that satisfies the cost upper and lower bounds for an arbitrary coalition *S* is identical to the *DID*.

We prove that the *DID* satisfies the non-cooperative core upper bounds for any coalition *S* by induction. The idea is that any coalition *S* can be created from the grand coalition *N* by consecutively deleting all non-members $m_j \in \{m_1, \ldots, m_z\}$ of *S* starting with the most downstream agent $m_z$. This procedure creates a sequence of intermediate coalitions $N = S_z, S_{z-1}, \ldots, S_1, S$. We show that the *DID* satisfies the core upper bounds for any intermediate coalition $S_j, j \in 1, \ldots, z$ and also for *S*.

For the grand coalition $N = S_z$, the non-cooperative core upper bounds are satisfied. Now, suppose the *DID* satisfies the non-cooperative core upper bounds for some intermediate coalition $S_j$, i.e.,

$$\sum_{i \in S_j} z_i^\star \leq v(S_j) . \tag{13}$$

We generate the intermediate coalition $S_{j-1}$ by deleting the non-member $m_j$ from the intermediate coalition $S_j$. By construction, the intermediate coalition $S_{j-1}$ consists of all strict predecessors of agent $m_j$ and all agents $i > m_j$ who belong to the coalition *S*. Rearranging inequality (13) and applying the definition of the *DID* implies

$$\sum_{i \in S_{j-1}} z_i^\star \leq v(S_j) - z_{m_j}^\star = v(S_j) - v(P_{m_j}) + v(P_{m_j} \backslash m_j) . \tag{14}$$

We have to show that the *DID* satisfies the non-cooperative core upper bounds for the intermediate coalition $S_{j-1}$, i.e.,

$$\sum_{i \in S_{j-1}} z_i^\star \leq v(S_j) - v(P_{m_j}) + v(P_{m_j} \backslash m_j) \leq v(S_{j-1}) . \tag{15}$$

Rearranging this inequality yields

$$v(S_j) - v(S_{j-1}) \leq v(P_{m_j}) - v(P_{m_j} \backslash m_j) . \tag{16}$$

If the coalition *S* does not have any members $i > m_j$, then the inequality is trivially satisfied as then $S_j = P_{m_j}$ and $S_{j-1} = P_{m_j} \backslash m_j$. Otherwise, define the set *T* consisting of all members *i* of the

coalition $S$ with $i > m_j$. Then, $S_j = P_{m_j} \cup T$ and $S_{j-1} = P_{m_j} \backslash m_j \cup T$ and by virtue of Proposition 3, inequality (16) holds.

We show the second part of the proof, by interpreting the cost upper bounds $a(S)$ as a game in characteristic function form on the set of all feasible coalitions $S$ of $N$.[4] Then, the downstream incremental distribution satisfies the upper cost bounds for any feasible coalition $S$ if and only if it is in the core of the game $a(S)$. In fact, the downstream incremental distribution is, by construction, a marginal contribution vector of the game $a(S)$. As shown by Shapley [14], for any convex game all marginal contribution vectors are in the core.[5] As Proposition 4 established that $a(S)$ is a convex game, the *DID* satisfies the cost lower bounds for any coalition $S$:

$$\sum_{i \in S} z_i^\star \geq a(S) . \tag{17}$$

Finally, we proof that the *DID* is the only river sharing agreement that simultaneously satisfies the cost upper and lower bounds for any coalition $S$. Therefore, we have to show that whenever a river sharing agreement $z$ satisfies both the cost upper and lower bounds, then for each agent $i$ it holds that $z_i = z_i^\star$. Again, the proof is by induction.

Similar to [9], for agent 1, any river sharing agreement $z$ fulfilling both constraints satisfies $v(\{1\}) \geq z_1 \geq a(\{1\})$. As $v(\{1\}) = a(\{1\})$ this implies $z_1 = z_1^\star$. Now, suppose that $z_i = z_i^\star$ holds for all agents $i$ upstream of some agent $j$, i.e., $i \leq j < n$. Summing up over all $i \in P_j$, we obtain

$$\sum_{i \in P_j} z_i = \sum_{i \in P_j} z_i^\star = v(P_j) . \tag{18}$$

As $v(P_{j+1}) = a(P_{j+1})$ and because any river sharing agreement $z$ satisfies both the cost upper and lower bounds, $\sum_{i \in P_{j+1}} z_i = v(P_{j+1}) = a(P_{j+1})$ has to hold. Hence,

$$z_{j+1} = \sum_{i \in P_{j+1}} z_i - \sum_{i \in P_j} z_i = v(P_{j+1}) - v(P_j) = z_{j+1}^\star. \tag{19}$$

Therefore, the cost distribution $z$ is identical to the *DID*. □

Theorem 1 is the exact counterpart to Theorem 1 of [9]. However, it is neither obvious nor straightforward to prove that the *DID* is the only distribution satisfying the cost upper and lower bounds in case of our river pollution model. The main challenge in [9] arose from the fact that cooperation among agents impose positive externalities on any coalition $S$. As a consequence, the welfare level a coalition could secure for itself crucially depends on the partition of all non-members. The same is true for our river pollution model. Cooperative behavior among non-members of a coalition $S$ induces, in general, positive abatement levels, which benefits the members of the coalition.

In contrast to [1,9], however, the decision variable in our model is emission abatement not water consumption. While water consumption only benefits the consumer and, thus, is a purely private commodity, emission abatement is not. In fact, in our model emission abatement does not benefit the abating agent but only all downstream agents, as it reduces the river's downstream pollution level. Thus, emission abatement imposes positive downstream externalities, i.e., pollution abatement is a commodity with public good properties. This is also reflected in the agents' welfare: agents' welfare in the water consumption models of [1,9] is simply given by some benefit function $b_i(x_i)$ which depends on the water consumption $x_i$ of agent $i$. In our model, the costs agent $i$ faces consist of two parts: first, the abatement costs $c_i(x_i)$, which only depend on the emission abatement of agent $i$ and, second,

---

[4]  We gratefully acknowledge that we owe this idea to an anonymous reviewer.
[5]  In fact, the core of a convex game lies within a convex polyhedron, the vertices of which are determined by the set of marginal contribution vectors, see [14].

the damage cost function $d_i(q_i)$ depending on the pollution level $q_i$, which itself is a function of the emission abatement levels of all upstream agents. In fact, ref. [9] is a special case of our model with $d_i(\cdot) = 0$ for all $i \in N$.

## 6. Discussion and Extensions

The model detailed in Section 2 relies on a number of assumptions which can be relaxed without impairing the statement of Theorem 1. First, we assumed that there is no initial pollution at the source of the river and that the net emissions of agent $i$ do not harm agent $i$ himself but only all downstream agents. As a consequence, agent 1 does not face any pollution and the specification of agent 1's damage function $d_1$ is optional. The first assumption simplified the specification of the pollution level $q_i$, while the latter assumption implied that in the non-cooperative Nash equilibrium no agent would abate at all. However, the proof of Theorem 1 does not draw on these assumptions and would still be valid if the pollution level agent $i$ faces was defined as

$$q_i = q_0 + \sum_{j \in P_i} \gamma_{ji}(e_j - x_j) \,, \tag{20}$$

where $q_0$ denotes an initial pollution level at the source of the river.

Second, we framed the model as a pollution abatement model. Obviously, emissions and the corresponding pollution levels are prime examples for downstream externalities, yet there are many other contexts to which our model is applicable. As an example, think of the case of flooding. Then, $e_i$ corresponds to the water discharges from the territory of agent $i$ into the river and $x_i$ denotes the amount of water agent $i$ withdraws from the stream (e.g., by the controlled flooding of designated flooding areas) and $q_i$ is the amount of excess water at agent $i$'s location. In this interpretation it would also be reasonable to assume that the water withdrawn $x_i$ is not limited by the discharge $e_i$ but could sum up to the total amount of excess water in the river basin, i.e.,

$$0 \le x_i \le q_i \,. \tag{21}$$

These modifications would also not impact on the validity of Theorem 1.

Third, particularly in case of flood protection, agents may have different means of protection. While the withdrawal of water induces costs to agent $i$ and benefits all agents further downstream, there are other protection techniques that are purely private goods. As an example, consider that agent $i$ could build a levee that protects the own territory from flooding, but does not induce any positive externalities to the downstream agents. Then, the damage to agent $i$ does not only depend on the total amount of water $q_i$ but also on the agent's investment into private damage protection $m_i$, i.e., $d_i = d_i(q_i, m_i)$. Assuming that an interior solution is optimal, i.e., $m_i^\star > 0$, the optimal level of private protection $m_i^\star(q_i)$ is given by the solution of the first order condition

$$\frac{\partial d_i(q_i, m_i)}{\partial m_i} = 0 \,. \tag{22}$$

Thus, we can re-write $d_i(q_i, m_i)$ as $d_i(q_i, m_i^\star(q_i))$. Whenever these newly specified damage functions $d_i(q_i, m_i^\star(q_i))$ are increasing, twice differentiable and convex in $q_i$, we are back at the model specification introduced in Section 2.

## 7. Conclusions

We showed that the main result of ref. [9], i.e., the downstream incremental distribution is the only welfare distribution satisfying the non-cooperative core bounds and the aspiration welfare bounds simultaneously, can be generalized to the case of commodities with public good characteristics. Like their water consumption model, our river pollution problem is a cooperative game with

externalities, since cooperation among non-members imposes a positive externality to the members of any coalition $S$. However, our model comprises an additional source of externalities, because the emissions discharged into the river induce negative externalities on all downstream agents. In addition, our results are robust to various extensions of our baseline model.

**Author Contributions:** Conceptualization, S.S. and R.W.; Formal analysis, S.S. and R.W.; Funding acquisition, R.W.; Investigation, S.S. and R.W.; Methodology, S.S. and R.W.; Validation, S.S. and R.W.; Visualization, S.S.; Writing—original draft, S.S. and R.W.; Writing—review & editing, R.W.

**Funding:** This research was funded by the Swiss National Science Foundation (SNF) grant number 154404.

**Acknowledgments:** We are grateful to Stefan Ambec, Eric Ansink, Hans-Peter Weikard, two anonymous reviewers and participants of the WCERE 2014 conference in Istanbul for valuable comments on an earlier draft.

**Conflicts of Interest:** The authors declare no conflict of interest. The funders had no role in the design of the study; in the collection, analyses, or interpretation of data; in the writing of the manuscript, or in the decision to publish the results.

## Appendix A. Proof of Proposition 3

To prove Proposition 3, set $S_j = P_{m_j} \cup T$ and $S_{j-1} = P_{m_j \setminus m_j} \cup T$. Let us parameterize the damage functions for agents $j > m_j$ with a parameter $\alpha \in [0, \infty)$. Due to this parametrization, the secure costs $v(S_{j-1}, \alpha)$ of the intermediate coalitions $S_{j-1}$ now depend on the parameter $\alpha$:

$$
\begin{aligned}
v(S_{j-1}, \alpha) = \sum_{i \in P_{m_j} \setminus m_j} k_i(x^v(S_{j-1}, \alpha)) + \sum_{i \in F_{m_j} \setminus m_j \cap S} c_i(x^v(S_{j-1}, \alpha)) \\
+ \alpha \sum_{i \in F_{m_j} \setminus m_j \cap S} d_i(q_i(x^v(S_{j-1}, \alpha))) .
\end{aligned}
\tag{A1}
$$

Then, inequality (8) can be generalized to

$$
v(S_j, \alpha) - v(S_{j-1}, \alpha) \leq v(P_{m_j}) - v(P_{m_j} \setminus m_j) , \quad \forall \alpha \in [0, \infty) .
\tag{A2}
$$

We now show that inequality (A2) holds for all $\alpha \in [0, \infty]$. For $\alpha = 0$, we have $v(S_j, 0) = v(P_{m_j})$ and $v(S_{j-1}, 0) = v(P_{m_j} \setminus m_j)$, and inequality (A2) holds with equality. For all other $\alpha$, we differentiate inequality (A2) with respect to $\alpha$:

$$
\frac{dv(S_j, \alpha)}{d\alpha} - \frac{dv(S_{j-1}, \alpha)}{d\alpha} \leq 0 .
\tag{A3}
$$

Applying the envelope theorem, we obtain

$$
\begin{aligned}
\frac{dv(S_j, \alpha)}{d\alpha} &= \frac{\partial v(x^v(S_j, \alpha), \alpha)}{\partial \alpha} + \underbrace{\frac{\partial v(x^v(S_j, \alpha), \alpha)}{\partial x(S_j, \alpha)} \frac{\partial x(S_j, \alpha)}{\partial \alpha}}_{0} \\
&= \frac{\partial v(x^v(S_j, \alpha), \alpha)}{\partial \alpha} = \sum_{i \in F_{m_j} \setminus m_j \cap S_j} d_i(q_i(x^v(S_j, \alpha))) ,
\end{aligned}
\tag{A4}
$$

and, analogously,

$$
\frac{dv(S_{j-1}, \alpha)}{d\alpha} = \frac{\partial v(x^v(S_{j-1}, \alpha), \alpha)}{\partial \alpha} = \sum_{i \in F_{m_j} \setminus m_j \cap S_{j-1}} d_i(q_i(x^v(S_{j-1}, \alpha))) .
\tag{A5}
$$

Then, inequality (A3) yields

$$\sum_{i \in F_{m_j} \setminus m_j \cap S_j} d_i(q_i(x^v(S_j), \alpha)) \leq \sum_{i \in F_{m_j} \setminus m_j \cap S_{j-1}} d_i(q_i(x^v(S_{j-1}, \alpha))) ,$$ (A6)

which is satisfied whenever

$$\sum_{l \in P_k} x_l^v(S_{j-1}, \alpha) \leq \sum_{l \in P_k} x_l^v(S_j, \alpha), \quad \forall k \in S_{j-1}, S_j .$$ (A7)

To see that this condition holds, consider two coalitions $S$ and $T = S \cup m$ with an agent $m \notin S$. Then, we obtain for inequality (A7):

$$\sum_{j \in P_k \cap S} x_j^v(S, \alpha) \leq \sum_{j \in P_k \cap T} x_j^v(T, \alpha), \quad \forall k \in S, T .$$ (A8)

To see that this holds, suppose the opposite:

$$\sum_{j \in P_k \cap T} x_l^v(T, \alpha) < \sum_{j \in P_k \cap S} x_l^v(S, \alpha) .$$ (A9)

According to the parameterized minimization problem, the following first order conditions have to be satisfied

$$
\begin{aligned}
c_i'(x_i) &\leq \sum_{j \in F_i \setminus i \cap T \cap P_m} d_j'(q_j(x_j)) + \alpha \sum_{j \in F_m \setminus m \cap T} d_j'(q_j(x_j)) \\
&\leq \sum_{j \in F_i \setminus i \cap T \cap P_m} d_j' \left( \sum_{k \in P_j \setminus j} \gamma_{kj} e_k - \sum_{k \in P_j \setminus j \cap T} \gamma_{kj} x_k \right) \\
&\quad + \alpha \sum_{j \in F_m \setminus m \cap T} d_j' \left( \sum_{k \in P_j \setminus j} \gamma_{kj} e_k - \sum_{k \in P_j \setminus j \cap T} \gamma_{kj} x_k \right), \quad \forall i \in T ,
\end{aligned}
$$ (A10)

and

$$
\begin{aligned}
c_i'(x_i) &\leq \sum_{j \in F_i \setminus i \cap S \cap P_m \setminus m} d_j'(q_j(x_j)) + \alpha \sum_{j \in F_m \setminus m \cap S} d_j'(q_j(x_j)) \\
&\leq \sum_{j \in F_i \setminus i \cap S \cap P_m \setminus m} d_j' \left( \sum_{k \in P_j \setminus j} \gamma_{kj} e_k - \sum_{k \in P_j \setminus j \cap S} \gamma_{kj} x_k \right) \\
&\quad + \alpha \sum_{j \in F_m \setminus m \cap S} d_j' \left( \sum_{k \in P_j \setminus j} \gamma_{kj} e_k - \sum_{k \in P_j \setminus j \cap S} \gamma_{kj} x_k \right), \quad \forall i \in S .
\end{aligned}
$$ (A11)

Due to assumption (A9), the right hand side of (A10) for $i \in T$ is higher than the right hand side of (A11) for $i \in S$. This implies $c_i'(x_i(T, \alpha)) \geq c_i'(x_i(S, \alpha))$ for all agents $i \in S, T$ and thus, due to the characteristics of the cost function $c_i(.)$, $x_i^v(T, \alpha) \geq x_i^v(S, \alpha), \forall i$. This, however, implies

$$\sum_{j \in P_k \cap T} x_j(T, \alpha) > \sum_{j \in P_k \cap S} x_j(S, \alpha) ,$$ (A12)

which contradicts assumption (A9).

Thus, inequality (A2) is satisfied for all $\alpha \in [0, \infty]$ and, therefore, holds in particular for $\alpha = 1$ implying that inequality (8) holds.

## Appendix B. Proof of Proposition 4

To prove Proposition 4, we first show that for any two coalitions $S \subset T \subseteq N$ the following relationships hold for the abatement levels of an agent $j \in S, T$:

$$x_j^a(T \cup i) \geq x_j^a(S \cup i) \geq x_j^a(S) \quad \text{and} \quad x_j^a(T \cup i) \geq x_j^a(T) . \tag{A13}$$

It suffices to show that these inequalities hold for two coalitions $S$ and $T$ with $T = S \cup t, t \in N \backslash S$. Let us first establish that $x_j^a(G \cup i) \geq x_j^a(G)$ for all coalitions $G = T, S$. The first order conditions for an agent $j \in G$ respectively $j \in G \cup i$ read

$$c_j'(x_j) \leq \sum_{k \in F_j \backslash j \cap G \cup i} d_k' \left( \sum_{m \in P_k \backslash k \cap G \cup i} \gamma_{mk}(e_m - x_m) \right) , \tag{A14}$$

respectively

$$c_j'(x_j) \leq \sum_{k \in F_j \backslash j \cap G} d_k' \left( \sum_{m \in P_k \backslash k \cap G} \gamma_{mk}(e_m - x_m) \right) . \tag{A15}$$

The right hand side of the first order condition in (A14) is either larger than the right hand side of (A15), if $j \leq i$, or equal to it, if $j > i$. Thus, $x_j^a(G \cup i) \geq x_j^a(G)$, for all $j \in G, G \cup i$ and $G = T, S$. Due to $T \cup i = S \cup i \cup t$, it follows that $x_j^a(T \cup i) \geq x_j^a(S \cup i)$.

According to the definition of the cost lower bounds, we obtain

$$a(T \cup i) - a(T) = k_i(x_i^a(T \cup i)) + \sum_{j \in T} k_j(x_j^a(T \cup i)) - k_j(x_j^a(T))$$
$$= k_i(x_i^a(T \cup i)) + \sum_{j \in T \backslash S} k_j(x_j^a(T \cup i)) - k_j(x_j^a(T)) + \sum_{j \in S} k_j(x_j^a(T \cup i)) - k_j(x_j^a(T)) , \tag{A16}$$

and

$$a(S \cup i) - a(S) = k_i(x_i^a(S \cup i)) + \sum_{j \in S} k_j(x_j^a(S \cup i)) - k_j(x_j^a(S)) . \tag{A17}$$

Rearranging inequality (11) of Proposition 4, using the expressions above, yields

$$k_i(x_i^a(T \cup i)) - k_i(x_i^a(S \cup i)) + \sum_{j \in T \backslash S} k_j(x_j^a(T \cup i)) - k_j(x_j^a(T))$$
$$+ \sum_{j \in S} k_j(x_j^a(T \cup i)) - k_j(x_j^a(T)) + \sum_{j \in S} k_j(x_j^a(S)) - k_j(x_j^a(S \cup i)) \geq 0 . \tag{A18}$$

To show that inequality (A18) holds, we split the terms into three groups $I, II, III$:

$$\underbrace{\sum_{j \in S} k_j(x_j^a(T \cup i)) - k_j(x_j^a(T)) + \sum_{j \in S} k_j(x_j^a(S)) - k_j(x_j^a(S \cup i)) +}_{I}$$
$$\underbrace{\sum_{j \in T \backslash S} k_j(x_j^a(T \cup i)) - k_j(x_j^a(T))}_{II} + \underbrace{k_i(x_i^a(T \cup i)) - k_i(x_i^a(S \cup i))}_{III} \geq 0 . \tag{A19}$$

In the following, we establish that all three groups are non-negative, i.e., $I, II, III \geq 0$. First, we show that $I \geq 0$ holds:

$$\sum_{j \in S} k_j(x_j^a(T \cup i)) - k_j(x_j^a(T)) + \sum_{j \in S} k_j(x_j^a(S)) - k_j(x_j^a(S \cup i)) \geq 0 . \tag{A20}$$

To see this, we split (A20) into cost and damage functions:

$$\sum_{j \in S} c_j(x_j^a(T \cup i)) - c_j(x_j^a(T)) + c_j(x_j^a(S)) - c_j(x_j^a(S \cup i))$$
$$+ \sum_{j \in S} d_j(q_j(x_j^a(T \cup i))) - d_j(q_j(x_j^a(T))) + d_j(q_j(x_j^a(S))) - d_j(q_j(x_j^a(S \cup i))) \geq 0 . \tag{A21}$$

Then, inequality (A20) follows from the convexity of both the damage and cost functions of each agent $j \in S$ and from the relationships $x_j^a(T \cup i) \geq x_j^a(S \cup i) \geq x_j^a(S)$ and $x_j^a(T \cup i) \geq x_j^a(T)$ established in (A13). In fact, for each agent $j$ it holds that

$$c_j(x_j^a(T \cup i)) - c_j(x_j^a(S \cup i)) = m , \quad m \geq 0 ,$$
$$c_j(x_j^a(S)) - c_j(x_j^a(T)) = n , \quad n \leq 0 , \tag{A22}$$

with $|m| \geq |n|$ as illustrated in Figure A1.[6] Similarly,

$$d_j(q_j(x_j^a(T \cup i))) - d_j(q_j(x_j^a(S \cup i))) = m , \quad m \leq 0 ,$$
$$d_j(q_j(x_j^a(S))) - d_j(q_j(x_j^a(T))) = n , \quad n \geq 0 , \quad \forall j , \tag{A23}$$

with $|n| \geq |m|$ as illustrated in Figure A2. Thus, inequality (A20) holds.

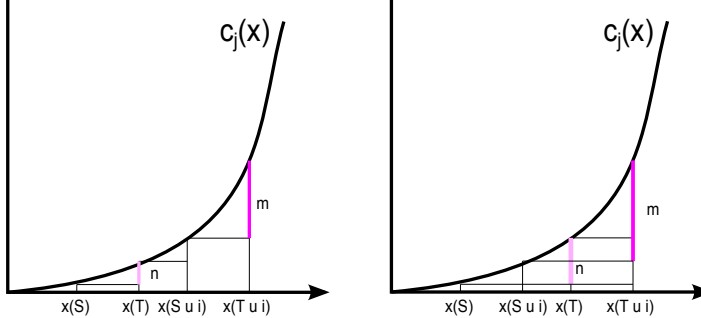

**Figure A1.** Cost functions.

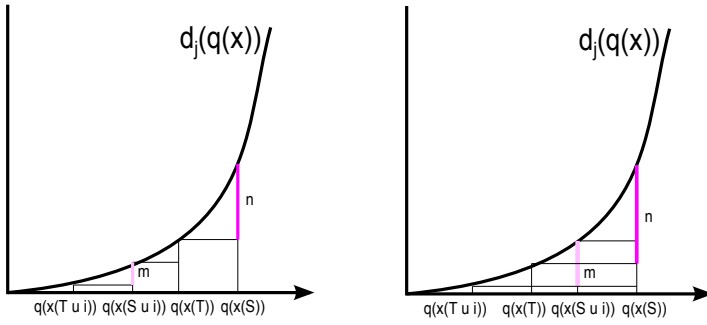

**Figure A2.** Damage cost functions.

Second, we show that $II \geq 0$ holds:

$$\sum_{j \in T \setminus S} k_j(x_j^a(T \cup i)) - k_j(x_j^a(T)) \geq 0 . \tag{A24}$$

---

[6]    As no general relationship can be established for $x_j^a(T)$ and $x^a(S \cup i)$, both cases are depicted in the Figures A1 and A2.

To see this, note that the following first order conditions hold

$$c_j'(x_j(T)) \le \sum_{k \in F_j \setminus j \cap T} d_k' \left( \sum_{t \in P_k \setminus k \cap T} \gamma_{tk}(e_t - x_t(T)) \right), \forall j \in T, \tag{A25}$$

and

$$c_j'(x_j(T \cup i)) \le$$
$$\sum_{k \in F_j \setminus j \cap T \cup i} d_k' \left( \sum_{t \in P_k \setminus k \cap T} \gamma_{tk} e_t + \gamma_{ik} e_i - \gamma_{tk} x_t(T) - \sum_{t \in P_k \setminus k \cap T \cup i} \gamma_{tk} \Delta x_t \right), \forall j \in T \cup i, \tag{A26}$$

with

$$\sum_{t \in P_k \setminus k \cap T \cup i} \gamma_{tk} \Delta x_t = \sum_{t \in P_k \setminus k \cap T \cup i} \gamma_{tk} x_j^a(T \cup i) - \sum_{t \in P_k \setminus k \cap T} \gamma_{tk} x_t^a(T). \tag{A27}$$

As $x_j^a(T \cup i) \ge x_j^a(T)$ as shown in (A13), the right hand side of inequality (A26) exceeds the right hand side of (A25). Thus, we obtain

$$\gamma_{ik} e_i \ge \sum_{t \in P_k \setminus k \cap T \cup i} \gamma_{tk} \Delta x_t = \sum_{t \in P_k \setminus k \cap T \cup i} \gamma_{tk} x_j^a(T \cup i) - \sum_{t \in P_k \setminus k \cap T} \gamma_{tk} x_t^a(T). \tag{A28}$$

Consequently,

$$d_j \left( \sum_{k \in P_j \setminus j \cap (T \cup i)} \gamma_{kj}(e_j - x_j^a(T \cup i)) \right) \ge d_j \left( \sum_{k \in P_j \setminus j \cap T} \gamma_{kj}(e_j - x_j^a(T)) \right), \forall j \in T. \tag{A29}$$

In addition, due to $x_j^a(T \cup i) \ge x_j^a(T)$ and $c_i(x_i)$ increasing and convex, we have $c_j(x_j^a(T \cup i)) \ge c_j(x_j^a(T))$. Together with (A29) this implies $k_j(x_j^a(T \cup i)) \ge k_j(x_j^a(T))$. Summing up over all $j \in T \setminus S$ yields inequality (A24).

Finally, we show that $III \ge 0$ holds:

$$k_i(x_i^a(T \cup i)) - k_i(x_i^a(S \cup i)) \ge 0. \tag{A30}$$

To see this, recall that the following first order condition holds for all $j \in S \cup i$

$$c_j'(x_j(S \cup i)) \le \sum_{k \in F_j \setminus j \cap S \cup i} d_k' \left( \sum_{t \in P_k \setminus k \cap S \cup i} \gamma_{tk}(e_t - x_t(S \cup i)) \right). \tag{A31}$$

Similarly, for each $j \in T \cup i$

$$c_j'(x_j(T \cup i)) \le \sum_{k \in F_j \setminus j \cap T \cup i} d_k' \Big( \sum_{t \in P_k \setminus k \cap S \cup i} \gamma_{tk} e_t + \sum_{t \in P_k \setminus k \cap T \setminus S} \gamma_{tk} e_t$$
$$- \sum_{t \in P_k \setminus k \cap S \cup i} \gamma_{tk} x_t(S \cup i) - \sum_{t \in P_k \setminus k \cap T \cup i} \gamma_{tk} \Delta x_t \Big). \tag{A32}$$

As $x_j^a(T \cup i) \ge x_j^a(S \cup i)$ due to (A13), the left hand side of inequality (A32) exceeds the left hand side of inequality (A31). Thus, it holds that

$$\sum_{t \in P_k \setminus k \cap T \setminus S} \gamma_{tk} e_t \ge \sum_{t \in P_k \setminus k \cap T \cup i} \gamma_{tk} \Delta x_t = \sum_{t \in P_k \setminus k \cap T \cup i} \gamma_{tk} x_t^a(T \cup i) - \sum_{t \in P_k \setminus k \cap S \cup i} \gamma_{tk} x_t^a(S \cup i). \tag{A33}$$

The agents $j \in T \cup i$ do not abate more than the additional pollution flow passing through their region compared to what they would optimally abate if they belong to the smaller coalition $S \cup i$. As a result,

$$d_i \left( \sum_{j \in P_i \setminus i \cap T \cup i} \gamma_{ji}(e_j - x_j^a(T \cup i)) \right) \geq d_i \left( \sum_{j \in P_i \setminus i \cap S \cup i} \gamma_{ji}(e_j - x_j^a(S \cup i)) \right) . \tag{A34}$$

In addition, as $x_j^a(T \cup i) \geq x_j^a(S \cup i)$ $\forall j$ due to (A13) and $c_i(x_i)$ is increasing and convex, we obtain

$$c_i(x_i^a(T \cup i)) \geq c_i(x_i^a(S \cup i)) . \tag{A35}$$

Combining (A34) and (A35) yields inequality (A30).

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
