# Peer review of "Sharing a River with Downstream Externalities"

_games, doi:10.3390/g10020023_

Round 1
Reviewer 1 Report
I am surprised the authors do not cite a paper by Gerard Laan and Nigel Moes published in NRM (2016), titled "COLLECTIVE DECISION MAKING IN AN INTERNATIONAL RIVER POLLUTION MODEL". This paper also takes a cooperative TU game to characterize solutions of a river polution game. In fact, one can argue that Laan & Moes is more general since they do not restrict themselves to the non-cooperative core (see current paper, line 158). As a result, Laan & Moes use a more general approach to tackle the same problem, and do not arrive at a unique characterization as the current paper does in Theorem 1.
Proposition 1 in the current paper is obvious since the authors assume that emissions only harm (strictly) downstream agents.
Proposition 2 is identical to proposition 1 of Laan & Moes (2016), albeit in terms of abatement, not polution.
Theorem 1 is the main contribution of this paper. It extends a result by Ambec & Sprumont (2002) to river sharing with externalities. I like it (but see my comment on Laan & Moes). This is a non-trivial contribution as the authors point out themselves.
Last sentence of Section 5: Perhaps replace this sentence by, again, stating that the model in the current paper generalizes Ambec & Sprumont and then stating the last sentence of Appendix C (which otherwise does not contribute much)?
One of the author names in ref [5] is spelled incorrectly.
Typos in appendix C: "there is a another" and "is a special case ours"
Author Response
Please, see attached file.

Reviewer 2 Report
My suggestions are listed in a pdf document

Author Response
Please, see attached file.

Round 2
Reviewer 1 Report
Dear authors,
Apologies for being tardy in looking at your revised manuscript. I have now taken a careful look. I appreciate your explanation of the differences between the current paper and Laan & Moes (2016). In fact, your explanation made me realize that, indeed, Laan & Moes take a strange detour. I agree with your observations and the way you describe the relation between both papers in section 1.
I am also satisfied with the other revisions made. Thanks.
Author Response
Dear reviewer,
thanks for your reply, which we highly appreciated.
Kind regards,
The authors
Reviewer 2 Report
Considering the new reference pointed out by the other reviewer, it seems to me that the comparison with van der Laan and Moes (2016) has to be improved. I have the feeling that te author's model can be written as in their article. This is perhaps something that might be formally establisehd. However, this does not call into question the merits of the article.
